# Fitness Gain of Individually Sensed Information by Cells

**DOI:** 10.3390/e21101002

**Published:** 2019-10-13

**Authors:** Tetsuya J. Kobayashi, Yuki Sughiyama

**Affiliations:** Institute of Industrial Science, The University of Tokyo, 4-6-1 Komaba, Meguro-ku, Tokyo 153-8505, Japan; yuki.sughiyama@gmail.com

**Keywords:** fluctuation theorem, evolution, decision-making, directed information, information thermodynamics, auto-encoder

## Abstract

Mutual information and its causal variant, directed information, have been widely used to quantitatively characterize the performance of biological sensing and information transduction. However, once coupled with selection in response to decision-making, the sensing signal could have more or less evolutionary value than its mutual or directed information. In this work, we show that an individually sensed signal always has a better fitness value, on average, than its mutual or directed information. The fitness gain, which satisfies fluctuation relations (FRs), is attributed to the selection of organisms in a population that obtain a better sensing signal by chance. A new quantity, similar to the coarse-grained entropy production in information thermodynamics, is introduced to quantify the total fitness gain from individual sensing, which also satisfies FRs. Using this quantity, the optimizing fitness gain of individual sensing is shown to be related to fidelity allocations for individual environmental histories. Our results are supplemented by numerical verifications of FRs, and a discussion on how this problem is linked to information encoding and decoding.

## 1. Introduction

Most biological systems are equipped with active sensing machinery to monitor the ever-changing environment. The fidelity of sensing is crucial to choosing appropriate states and behaviors in response to changes in environmental states [1,2,3]. Instantaneous mutual information, path-wise mutual information, and its causal variant, directed information, have been used to quantitatively characterize the performance of the sensing and information transduction, theoretically [4,5,6,7] and experimentally [8,9,10,11]. These information measures are also fundamental to the thermodynamic cost of sensing [12,13,14,15,16].

However, it is still elusive whether these measures can appropriately quantify the biological and fitness value of sensed information. Despite intensive works on the fitness value of information [17,18,19,20,21,22,23,24,25], almost all works considered a biologically unrealistic situation in which all cells or organisms in a population receive a common sensing signal, which is the requisite for proving that the fitness value of sensing is bounded by the information measures. Few studies have conjectured that biologically realistic sensing by individual organisms may have greater fitness value than these measures [22,25].

In this work, we resolve this problem by generally proving that the individual sensing always has greater fitness value than common sensing does. The additional fitness gain, which satisfies fluctuation relations (FRs), is attributed to the selection of organisms that obtains a correct sensing signal by chance. A new quantity, which is similar to the coarse-grained entropy production in information thermodynamics, is introduced to quantify the total fitness gain from the individual sensing, the upper bound of which is strictly higher than the directed information. We further show that the optimization of this quantity is closely related to optimizing an auto-encoding network, in which sensing, phenotypic switching, and metabolic allocation work as encoding, processing, and decoding, respectively. Our general results, especially those for FRs, are verified by a numerical simulations.

## 2. Modeling Sensing and Adaptation Processes

We consider a population of an asexual organism that replicates with an instantaneous replication rate k(x,y), depending on its phenotype $x\in S^{x}$ and the state of environment $y\in S^{y}$, where the phenotypic and environmental states are assumed to be discrete and finite, for simplicity. The organism switches its phenotype stochastically from *x* to $x^{\prime }$ by exploiting sensing signal $z\in S^{z}$ with a transition probability $T_{F}\left(x^{\prime }|x,z\right)$ within a small time interval Δt. Depending on the physical entity of *z*, the sensing can be categorized as either individual or common sensing [22,25]. In the case of individual sensing, *z* is the state of a sensing system of the organism, such as the activity of receptors. Due to the stochasticity in the sensing process, the individual organisms receive different sensing signals *z* (Figure 1a). By assuming that the stochastic sensing output *z* depends on the state of the environment *y* as $T_{S}\left(z^{\prime }|z,y^{\prime }\right)$, we describe the dynamics of the number of organisms $N_{t}^{Y}\left(x_{t},z_{t}\right)$ that have phenotypic state $x_{t}$ with sensing signal $z_{t}$ at *t* as
(1)$$\begin{array}{c} N_{t}^{Y}\left(x_{t+1},z_{t+1}\right)=e^{k(x_{t+1},y_{t+1})}\times \sum_{x_{t},z_{t}}T_{F}\left(x_{t+1}|x_{t},z_{t+1}\right)T_{S}\left(z_{t+1}|z_{t},y_{t+1}\right)N_{t}^{Y}\left(x_{t},z_{t}\right), \end{array}$$
where $Y_{t}:=\left\{y_{0},\cdots ,y_{t}\right\}$ is the history of the environmental state, the statistical properties of which are characterized by a path probability $Q[Y_{t}]$. In this representation, we implicitly assume a causal dependency among $x_{t}$, $y_{t}$, and $z_{t}$ as $\cdots \to y_{t}\to z_{t}\to x_{t}\to y_{t+1}\to z_{t+1}\to x_{t+1}\to \cdots$. Such representation has been conventionally used for notational simplicity.

In contrast, in the case of common sensing, *z* is assumed to be partial information on the environmental state that is common to all organisms [26,27] (Figure 1b). An example is an extracellular chemical that correlates with the environmental state and can be sensed by the organisms with negligible error. The dynamics of the number of organisms $N_{t}^{Y,Z}\left(x_{t}\right)$ with phenotypic state *x* at time *t* under a realization of environmental and common signal histories, $Y_{t}$ and $Z_{t}$, can be represented as

(2)$$\begin{array}{cc} N_{t+1}^{Y,Z}\left(x_{t+1}\right)= & e^{k(x_{t+1},y_{t+1})}\times \sum_{x_{t}\in S^{x}}T_{F}\left(x_{t+1}|x_{t},z_{t+1}\right)N_{t}^{Y,Z}\left(x_{t}\right). \end{array}$$

We assume that the history of the common signal $Z_{t}:=\left\{z_{0},\cdots ,z_{t}\right\}$ follows a statistical law $Q[Z_{t}∥Y_{t}]$, which is causally conditional on the environmental history. At this stage, the statistical property of the common signal is abstractly represented as $Q[Z_{t}∥Y_{t}]$. However, in the following, we are going to assume that $Q[Z_{t}∥Y_{t}]$ is identical to the path probability generated by the sensing law of the individual sensing, $T_{S}\left(z^{\prime }|z,y^{\prime }\right)$, to clarify the difference of individual and common sensing. While common sensing is not biologically realistic enough, most previous works on the fitness value of information only addressed common sensing, and proved that the fitness gain of common sensing is upper bounded by the directed information [26,27].

Before deriving relations between fitness gain and information measures, we will mention the limitations of the model we assumed. In our modeling above, we did not include the carrying capacity of the environment, which works to reduce the growth of individual cells when the number of cells in the environment approaches the capacity. Additionally, we also assumed that cells cannot affect the behavior of the environment. Even though these factors are biologically important and also theoretically intriguing, we did not assume them, as the following information-theoretic analysis of the population dynamics cannot be applicable at this moment. We touch on potential extensions of this work to include these factors in our Discussion.

### Fitness of a Population with Individual and Common Sensing

The fitness of a population with individual sensing $\Psi^{i}\left[Y_{t}\right]$ and with common sensing $\Psi^{c}\left[Y_{t},Z_{t}\right]$ can be defined respectively as
(3)$$\begin{array}{c} \Psi^{i}\left[Y_{t}\right]:=\ln\frac{N_{t}^{Y}}{N_{0}^{Y}},\;\Psi^{c}\left[Y_{t},Z_{t}\right]:=\ln\frac{N_{t}^{Y,Z}}{N_{0}^{Y,Z}}, \end{array}$$
where $N_{t}^{Y}:=\sum_{x_{t},z_{t}}N_{t}^{Y}\left(x_{t},z_{t}\right)$ and $N_{t}^{Y,Z}:=\sum_{x_{t}}N_{t}^{Y,Z}\left(x_{t}\right)$. We define a pathwise historical fitness [28]
(4)$$\begin{array}{c} K\left[X_{t},Y_{t}\right]:=\sum_{\tau =0}^{t-1}k\left(x_{\tau +1},y_{\tau +1}\right), \end{array}$$
and path probabilities for phenotypic and signal histories
(5)$$\begin{array}{cc} P_{F}\left[X_{t}∥Z_{t}\right] & :=\left[\prod_{\tau =0}^{t-1}T_{F}\left(x_{\tau +1}|x_{\tau },z_{\tau +1}\right)\right]p_{F}\left(x_{0}|z_{0}\right), \end{array}$$
(6)$$\begin{array}{cc} P_{S}\left[Z_{t}∥Y_{t}\right] & :=\left[\prod_{\tau =0}^{t-1}T_{S}\left(z_{\tau +1}|z_{\tau },y_{\tau +1}\right)\right]p_{S}\left(z_{0}|y_{0}\right), \end{array}$$
respectively. Then, by using Equations (Equation 1) and (Equation 2), we can explicitly represent the fitnesses [26,27,28,29] as
(7)$$\begin{array}{cc} \Psi^{i}\left[Y_{t}\right]= & \ln{\langle e^{K[X_{t},Y_{t}]}\rangle }_{P_{F,S}\left[X_{t}|Y_{t}\right]}, \end{array}$$
(8)$$\begin{array}{cc} \Psi^{c}\left[Y_{t},Z_{t}\right]= & \ln{\langle e^{K[X_{t},Y_{t}]}\rangle }_{P_{F}\left[X_{t}∥Z_{t}\right]}, \end{array}$$
where ${\langle \cdot \rangle }_{P[X_{t}]}$ is the average with respect to $P[X_{t}]$, and $P_{F,S}\left[X_{t}|Y_{t}\right]:=\sum_{Z_{t}}P_{F}\left[X_{t}∥Z_{t}\right]P_{S}\left[Z_{t}∥Y_{t}\right]$ (see also Section A.1 for the derivation). Here, ∥ is the Kramer’s causal conditioning, which indicate a causal relation between the conditioning and the conditioned histories [30,31]. Using the path representation of the fitnesses, we can define the time-backward retrospective path probabilities as
(9)$$\begin{array}{cc} P_{B}^{i}\left[X_{t},Z_{t}|Y_{t}\right] & :=\frac{N_{t}^{Y}\left[X_{t},Z_{t},Y_{t}\right]}{N_{t}^{Y}}=e^{K\left[X_{t},Y_{t}\right]-\Psi^{i}\left[Y_{t}\right]}P_{F}\left[X_{t}∥Z_{t}\right]P_{S}\left[Z_{t}∥Y_{t}\right], \end{array}$$
(10)$$\begin{array}{cc} P_{B}^{c}\left[X_{t}|Y_{t},Z_{t}\right] & :=\frac{N_{t}^{Y,Z}\left[X_{t},Z_{t},Y_{t}\right]}{N_{t}^{Y,Z}}=e^{K\left[X_{t},Y_{t}\right]-\Psi^{c}\left[Y_{t},Z_{t}\right]}P_{F}\left[X_{t}∥Z_{t}\right],, \end{array}$$
where $N_{t}^{Y}\left[X_{t},Z_{t},Y_{t}\right]$ is the number of cells at time *t* that have phenotypic and individual sensing histories, $X_{t}$ and $Z_{t}$, under the realization of environmental history $Y_{t}$. Similarly, $N_{t}^{Y,Z}\left[X_{t},Z_{t},Y_{t}\right]$ is the number of cells at time *t* that have phenotypic history, $X_{t}$, under the realization of environmental and common sensing histories $Y_{t}$ and $Z_{t}$. Thus, $P_{B}^{i}$ and $P_{B}^{c}$ can be interpreted as the probabilities of observing a phenotypic history $X_{t}$ when we trace the phenotypic history from time *t* to time 0 in a time-backward manner, retrospectively [26,27,29]. In contrast, $P_{F}\left[X_{t}∥Z_{t}\right]$ is the probability of observing $X_{t}$ when we trace the phenotypic history in a time forward manner [26,27,29]. The difference between the two is attributed to the impact of selection, which can be characterized by investigating a population after selection, retrospectively.

## 3. Stochastic Trajectories of Individual and Common Sensing

In order to provide numerical examples of the difference between individual and common sensing, we consider a Markovian environment with three states, $S^{y}=\left\{s_{1}^{y},s_{2}^{y},s_{3}^{y}\right\}$, and a population with two phenotypic states, $S^{x}=\left\{s_{1}^{x},s_{2}^{x}\right\}$. Of the three environmental states, $s_{1}^{y}$ and $s_{2}^{y}$ are nutrient A- and nutrient B-rich environments, respectively. The environmental states fluctuate between these two states, most of time (Figure 2a). In contrast, $s_{3}^{y}$ is a nutrient-poor environment, in which the growth of the population is limited (Figure 2b). The environmental state occasionally sojourns in this state $s_{3}^{y}$ from either $s_{1}^{y}$ or $s_{2}^{y}$ (Figure 2a). The rule for these stochastic transitions among the environmental states is specified by a stochastic transition matrix, $T_{E}^{F}\left(y^{\prime }|y\right)$, from *y* to $y^{\prime }$:(11)$$\left\{T_{E}^{F}\left(y^{\prime }|y\right)\right\}=\begin{array}{cc} & \begin{array}{ccc} s_{1}^{y} & s_{2}^{y} & s_{3}^{y} \end{array}\\ \begin{array}{c} s_{1}^{y}\\ s_{2}^{y}\\ s_{3}^{y} \end{array} & \left(\begin{array}{ccc} 0.70 & 0.25 & 0.25\\ 0.25 & 0.70 & 0.25\\ 0.05 & 0.05 & 0.50 \end{array}\right) \end{array}.$$

The two phenotypic states, $s_{1}^{x}$ and $s_{2}^{x}$, are assumed to be adapted specifically to the nutrient A-rich state $s_{1}^{y}$ and the nutrient B-rich state $s_{2}^{y}$, respectively. These are modeled by the replication rates $k(s_{1}^{x},s_{1}^{y})$ and $k(s_{2}^{x},s_{2}^{y})$ in the adaptive environments, which are higher than those of $k(s_{1}^{x},s_{2}^{y})$ and $k(s_{2}^{x},s_{1}^{y})$ in the non-adaptive environment (Figure 2b):(12)$$\left\{e^{k(x,y)}\right\}=\begin{array}{cc} & \begin{array}{ccc} s_{1}^{y} & s_{2}^{y} & s_{3}^{y} \end{array}\\ \begin{array}{c} s_{1}^{x}\\ s_{2}^{x} \end{array} & \left(\begin{array}{ccc} 2.24 & 0.32 & 0.08\\ 0.32 & 2.24 & 0.08 \end{array}\right) \end{array}.$$

The sensing signal has two states, $S^{z}=\left\{s_{1}^{z},s_{2}^{z}\right\}$, which correspond to the nutrient A- and nutrient B-rich environments, $s_{1}^{y}$ and $s_{2}^{y}$, respectively. A cell in the case of individual sensing, or cells in the case of the common sensing, receive $s_{1}^{z}$ or $s_{2}^{z}$ with high probability when the environmental state is $s_{1}^{y}$ or $s_{2}^{y}$, respectively. If the environment is in the nutrient-poor $s_{3}^{y}$ state, a cell or cells obtain $s_{1}^{z}$ or $s_{2}^{z}$ with equal probability. Here, the sensing is assumed to be memory-less as $T_{S}\left(z^{\prime }|z,y^{\prime }\right)=T_{S}\left(z^{\prime }|y^{\prime }\right)$, and, thus, its stochastic behavior is defined by a transition matrix, $T_{S}\left(z|y\right)$, for individual sensing, and by $T_{E}^{F}\left(z|y\right)$ for common sensing (Figure 2c):(13)$$\left\{T_{S}\left(z|y\right)\right\}=\left\{T_{E}^{F}\left(z|y\right)\right\}=\begin{array}{cc} & \begin{array}{ccc} s_{1}^{y} & s_{2}^{y} & s_{3}^{y} \end{array}\\ \begin{array}{c} s_{1}^{z}\\ s_{2}^{z} \end{array} & \left(\begin{array}{ccc} 0.8 & 0.2 & 0.5\\ 0.2 & 0.8 & 0.5 \end{array}\right) \end{array}.$$

In order to compare individual and common sensing, we set the accuracy of sensing to be equal, $T_{S}\left(z|y\right)=T_{E}^{F}\left(z|y\right)$, for all $y\in S^{y}$ and $z\in S^{z}$. Finally, a cell is assumed to switch into phenotypic state $s_{i}^{x}$ with high probability when it receives a sensing signal $s_{i}^{z}$ for i={1,2} (Figure 2d):(14)$$\left\{T_{F}\left(x^{\prime }|z\right)\right\}=\begin{array}{cc} & \begin{array}{cc} s_{1}^{z} & s_{2}^{z} \end{array}\\ \begin{array}{c} s_{1}^{x}\\ s_{2}^{x} \end{array} & \left(\begin{array}{cc} 0.95 & 0.05\\ 0.05 & 0.95 \end{array}\right) \end{array},$$
where the phenotypic switching is set to be memory-less $T_{F}\left(x^{\prime }|x,z\right)=T_{F}\left(x^{\prime }|z\right)$.

Given these conditions, Figure 3 illustrates the population dynamics of cells with individual sensing Figure 3a,b and with common sensing Figure 3c,d under two different realizations of the environment. For the first realization, shown in Figure 3a,c,e, $\Psi^{i}\left[Y_{t}\right]$ is higher than $\Psi^{c}\left[Y_{t},Z_{t}\right]$ (see red and blue solid lines in Figure 3e), whereas, for the second realization (Figure 3b,d,f)), $\Psi^{c}\left[Y_{t},Z_{t}\right]$ is greater than $\Psi^{i}\left[Y_{t}\right]$ (Figure 3f). This clearly illustrates that the fitness advantages of individual and common sensing are strongly dependent on the actual realization of the environment and the common sensing signal. When common sensing produces a correct signal by chance, the population with common sensing can enjoy a higher fitness gain than that with individual sensing. By contrast, the population with common sensing loses fitness when the signal is incorrect. Figure 4 also shows the behaviors of $\Psi^{i}\left[Y_{t}\right]$ (Figure 4b) and $\Psi^{c}\left[Y_{t},Z_{t}\right]$ (Figure 4c) under 100 different realizations of $\left\{Y_{t},Z_{t}\right\}$, which reinforces the observation that both $\Psi^{i}\left[Y_{t}\right]$ and $\Psi^{c}\left[Y_{t},Z_{t}\right]$ can fluctuate significantly, depending on the realizations. However, an ensemble average of the fitness shows that ${\langle \Psi^{i}\rangle }_{Q}$ is greater than ${\langle \Psi^{c}\rangle }_{Q}$, at least for this specific instance (the red and blue solid lines in Figure 4a).

## 4. Value of Individual Sensing is Always Greater than that of Common Sensing

In order to characterize the fitness difference between individual and common sensing in general, we derive a detailed fluctuation relation for the fitness difference $g\left[Y_{t},Z_{t}\right]:=\Psi^{i}\left[Y_{t}\right]-\Psi^{c}\left[Y_{t},Z_{t}\right]$ from Equations (Equation 9) and (10) as
(15)$$\begin{array}{c} e^{-g[Y_{t},Z_{t}]}=\frac{e^{\Psi^{c}\left[Y_{t},Z_{t}\right]}}{e^{\Psi^{i}\left[Y_{t}\right]}}=\frac{P_{B}^{i}\left[X_{t},Z_{t}|Y_{t}\right]}{P_{B}^{c}\left[X_{t}|Y_{t},Z_{t}\right]P_{S}\left[Z_{t}∥Y_{t}\right]}=\frac{P_{B}^{i}\left[Z_{t}|Y_{t}\right]}{P_{S}\left[Z_{t}∥Y_{t}\right]}, \end{array}$$
where $P_{B}^{i}\left[Z_{t}|Y_{t}\right]:=\sum_{X_{t}}P_{B}^{i}\left[X_{t},Z_{t}|Y_{t}\right]$ (see also Section A.2 for the derivation). By assuming that the statistical property of common sensing is the same as that of individual sensing, $Q\left[Z_{t}∥Y_{t}\right]=P_{S}\left[Z_{t}∥Y_{t}\right]$, as in Figure 3 and Figure 4, we obtain the average fluctuation relation (FR) as
(16)$$\begin{array}{c} {\langle \Psi^{i}\left[Y_{t}\right]\rangle }_{Q[Y_{t}]}-{\langle \Psi^{c}\left[Y_{t},Z_{t}\right]\rangle }_{Q[Y_{t},Z_{t}]}=G, \end{array}$$
where
(17)$$\begin{array}{c} G:={\langle g\rangle }_{Q}=D\left[P_{S}\left[Z_{t}∥Y_{t}\right]Q\left[Y_{t}\right]∥P_{B}^{i}\left[Z_{t}|Y_{t}\right]Q\left[Y_{t}\right]\right], \end{array}$$
is the Kulback–Leibler (KL) divergence between the time-forward sensing behavior, $P_{S}\left[Z_{t}∥Y_{t}\right]$, and the time-backward behavior, $P_{B}^{i}\left[Z_{t}|Y_{t}\right]$ (see also Section A.3 for derivation). Together with the non-negativity of the KL divergence, the average FR indicates that the average fitness of individual sensing is always greater than that of common sensing by G≥0. As individual and common sensing are assumed to have the same statistical property, the source of the gain G is attributed to the individuality of the sensing. In the case of individual sensing, the organisms receiving the correct signal by chance grow more than those that receive incorrect signal do. Thus, the retrospective signal histories $P_{B}^{i}\left[Z_{t}|Y_{t}\right]$ are biased by the selection from the time-forward signal histories $P_{S}\left[Z_{t}∥Y_{t}\right]$. The gain G is exactly this bias, quantified by the KL divergence. No such gain is obtained from the common sensing, because the sensing signal is common to all organisms and, thus, no bias is induced by selection. This result clearly indicates that the fitness value of individual sensing cannot be properly evaluated by considering only the time-forward behavior of the signal and the environment. Even though individual sensing gains more fitness than common sensing does, on average, as demonstrated in Figure 4a, $g[Y_{t},Z_{t}]$ fluctuates significantly and common sensing can gain more fitness than individual sensing does, by chance (Figure 3b,d and Figure 5a). From the detailed FR for $g[Y_{t},Z_{t}]$ Equation (Equation 15), we also derive the integral fluctuation relation (IFR):$$\begin{array}{c} {\langle e^{-g[Y_{t},Z_{t}]}\rangle }_{Q[Y_{t},Z_{t}]}={\langle e^{-(\Psi^{i}\left[Y_{t}\right]-\Psi^{c}\left[Y_{t},Z_{t}\right])}\rangle }_{Q[Y_{t},Z_{t}]}=1, \end{array}$$
which clarifies that $g[Y_{t},Z_{t}]$ fluctuates, such that the positive $g[Y_{t},Z_{t}]$ balances the negative $g[Y_{t},Z_{t}]$ to satisfy the equality. The integral FR is also verified numerically in Figure 5a,b.

### 4.1. The Gain of Fitness by Individual Sensing

We further investigate $\Psi^{i}\left[Y_{t}\right]$ to clarify how the fitness of the organisms with individual sensing is shaped. To this end, as in a previous work [27], which investigated the fitness value of common sensing, we additionally assume that k(x,y) can be decomposed as $e^{k(x,y)}=e^{k_{max}\left(y\right)}T_{K}\left(y|x\right)$ [27]. In this decomposition, $k_{max}\left(y\right)$ can be interpreted as the maximum replication rate that would be attained if the organisms allocated all their metabolic resources to adapt only to the environmental state *y*. Therefore, under this extreme allocation, the organisms die out under the environmental states other than *y*. The decomposition effectively means that we presume that each phonotypic state can be characterized by how a cell allocates its metabolic resources to different environmental conditions. $T_{K}\left(y|x\right)$ is the fraction of metabolic resources allocated to the environmental state *y* in a phenotypic state *x*. Thereby, we call $T_{K}\left(y|x\right)$ metabolic allocation strategy of the organisms in this work. The biological motivation behind the metabolic allocation is the problem of generalist and specialist. In order to adapt to a changing environment, a cell has essentially two tactics: One is to equip a cell with single fixed phenotypic state that distributes the metabolic resources over all the possible environmental states. Thereby, such a cell can, at least, evade extinction under any environmental state. The other is to switch between multiple specialized phenotypic states, each of which allocates the metabolic resources to a small number of environmental states. In this case, each phenotypic state runs the risk of extinction, but such risk is hedged as a population by stochastic switching of the phenotypic states. These two are continuously interpolated, and the optimal one depends on the way the environment fluctuates. The introduction of the metabolic allocation strategy enables us to consider a wider spectrum of biological adaption in which the character of each phenotypic state is also optimized evolutionarily.

By defining
(18)$$\begin{array}{cc} P_{K}\left[Y_{t}∥X_{t}\right] & :=\prod_{\tau =0}^{t-1}T_{K}\left(y_{\tau +1}|x_{\tau +1}\right), \end{array}$$
(19)$$\begin{array}{cc} K_{\max}\left[Y_{t}\right] & :=\sum_{\tau =1}^{t}k_{\max}\left(y_{\tau }\right), \end{array}$$
the historical fitness, Equation (Equation 4), is decomposed as

(20)$$\begin{array}{c} K\left[X_{t},Y_{t}\right]=K_{\max}\left[Y_{t}\right]+\ln P_{K}\left[Y_{t}∥X_{t}\right]. \end{array}$$

By introducing this decomposition into Equation (Equation 9), we obtain
(21)$$\begin{array}{c} e^{\Psi^{i}\left[Y_{t}\right]-\Psi_{0}\left[Y_{t}\right]}=\frac{P_{K}\left[Y_{t}∥X_{t}\right]P_{F}\left[X_{t}∥Z_{t}\right]P_{S}\left[Z_{t}∥Y_{t}\right]}{P_{B}^{i}\left[X_{t},Z_{t}|Y_{t}\right]Q\left[Y_{t}\right]}, \end{array}$$
where $\Psi_{0}\left[Y_{t}\right]:=K_{\max}\left[Y_{t}\right]+\ln Q\left[Y_{t}\right]$, the average of which is known to bound the average fitness of a population without sensing [27] (see Section A.4 for derivation). By taking the marginalization with respect to $X_{t}$ and $Z_{t}$, we have
(22)$$\begin{array}{c} \Psi^{i}\left[Y_{t}\right]=\Psi_{0}\left[Y_{t}\right]+\sigma \left[Y_{t}\right]=K_{\max}\left[Y_{t}\right]+\ln P_{KFS}\left[Y_{t}|Y_{t}\right], \end{array}$$
where
$$\begin{array}{cc} P_{KFS}\left[Y_{t}^{\prime }|Y_{t}\right] & :=\sum_{X_{t},Z_{t}}P_{K}\left[Y_{t}^{\prime }∥X_{t}\right]P_{F}\left[X_{t}∥Z_{t}\right]P_{S}\left[Z_{t}∥Y_{t}\right], \end{array}$$
and
$$\begin{array}{cc} \sigma [Y_{t}] & :=\ln\frac{P_{KFS}\left[Y_{t}|Y_{t}\right]}{Q[Y_{t}]}. \end{array}$$

Since the average of $\Psi_{0}\left[Y_{t}\right]$ is the tight bound of the fitness without sensing, $\sigma [Y_{t}]$ is the gain of fitness by individual sensing. Here, $P_{KFS}\left[Y_{t}^{\prime }|Y_{t}\right]$ is the probability that an organism allocates its metabolic resources to an environmental history $Y_{t}^{\prime }$ when it experiences environmental history $Y_{t}$. Thus, $P_{KFS}\left[Y_{t}|Y_{t}\right]$ measures the probability that the metabolic resource is correctly allocated to the actual environmental history $Y_{t}$, and $1-P_{KFS}\left[Y_{t}|Y_{t}\right]$ is the probability of an incorrect allocation. In other words, $P_{KFS}\left[Y_{t}|Y_{t}\right]$ characterizes how accurately the individual sensing, phenotypic switching, and metabolic allocation together respond to the actual environment. From an information-theoretic viewpoint, this cascade from environment to metabolic allocation via sensing and phenotypic switching is very similar to the auto-encoding and decoding of information $Y_{t}$ via multiple layers [32]. The sensing works as the encoding of an environmental history $Y_{t}$ into $Z_{t}$. The signal-dependent phenotypic switching is the processing of the encoded signal in the internal layers. The metabolic allocation is the decoding process to recover the original information, $Y_{t}$, from $X_{t}$. Under this interpretation, $P_{KFS}\left[Y_{t}^{\prime }|Y_{t}\right]$ determines the statistical correspondence between the encoded information $Y_{t}$ and the decoded information $Y_{t}^{\prime }$, and $P_{KFS}\left[Y_{t}|Y_{t}\right]$ is the probability that the encoded data $Y_{t}$ is correctly decoded as $Y_{t}$. Therefore, the total fidelity can be quantified as

(23)$$\begin{array}{c} \gamma_{t}:=\ln\sum_{Y_{t}}P_{KFS}\left[Y_{t}|Y_{t}\right]=\ln{\langle e^{\sigma [Y_{t}]}\rangle }_{Q[Y_{t}]}. \end{array}$$

Formally, similar quantities to $\sigma [Y_{t}]$ and $\gamma_{t}$ were introduced by Sagawa and Ueda as the coarse-grained entropy production and the efficiency parameter of feedback control in information thermodynamics [33,34]. Using $\gamma_{t}$, $\sigma [Y_{t}]$ can be decomposed as
$$\begin{array}{c} \sigma \left[Y_{t}\right]=\gamma_{t}-\ln\frac{Q[Y_{t}]}{P_{\gamma }\left[Y_{t}\right]}, \end{array}$$
where
(24)$$\begin{array}{c} P_{\gamma }\left[Y_{t}\right]:=e^{-\gamma_{t}}P_{KFS}\left[Y_{t}|Y_{t}\right], \end{array}$$
is a path probability. By combining this with Equation (Equation 22), we have
(25)$$\begin{array}{c} \Psi^{i}\left[Y_{t}\right]=\Psi_{0}\left[Y_{t}\right]+\gamma_{t}-\ln\frac{Q[Y_{t}]}{P_{\gamma }\left[Y_{t}\right]}. \end{array}$$

By taking the average with respect to $Q[Y_{t}]$, we obtain

(26)$$\begin{array}{c} {\langle \Psi^{i}\rangle }_{Q}={\langle \Psi_{0}\rangle }_{Q}+\gamma_{t}-D\left[Q\left[Y_{t}\right]∥P_{\gamma }\left[Y_{t}\right]\right]\leq {\langle \Psi_{0}\rangle }_{Q}+\gamma_{t}. \end{array}$$

Equations (Equation 25) and (Equation 26) can be regarded as detailed and average FRs, respectively, with respect to $\Psi_{0}\left[Y_{t}\right]+\gamma_{t}-\Psi^{i}\left[Y_{t}\right]$. As ${\langle \Psi_{0}\rangle }_{Q}$ is the tight upper bound of the average fitness without sensing, this relation means that $\gamma_{t}$ is an upper bound of the fitness gain from individual sensing. Moreover, $\gamma_{t}$ is an intrinsic quantity of the population, in the sense that it is determined irrespective of the actual statistical law of the environment, $Q[Y_{t}]$. The deviation of $\Psi^{i}\left[Y_{t}\right]$ from ${\langle \Psi_{0}\rangle }_{Q}+\gamma_{t}$ satisfies an integral FR as
(27)$$\begin{array}{c} {\langle e^{-(\Psi_{0}\left[Y_{t}\right]+\gamma_{t}-\Psi^{i}\left[Y_{t}\right])}\rangle }_{Q[Y_{t}]}={\langle e^{-(\gamma_{t}-\sigma \left[Y_{t}\right])}\rangle }_{Q[Y_{t}]}=1, \end{array}$$
the behaviors of which are illustrated numerically in Figure 5c,d.

### 4.2. Connection with Other Information Measures

In order to link the quantities σ and $\gamma_{t}$ with other common information measures, we further assume that the environment is Markovian:(28)$$\begin{array}{c} Q\left[Y_{t}\right]=\prod_{\tau =0}^{t-1}T_{E}^{F}\left(y_{\tau +1}|y_{\tau }\right)p_{E}\left(y_{0}\right), \end{array}$$
and that the sensing is memory less as

(29)$$\begin{array}{c} T_{S}\left(z_{t+1}|Z_{t},y_{t+1}\right)=T_{S}\left(z_{t+1}|y_{t+1}\right). \end{array}$$

Then, we obtain the joint time-forward probability for $Y_{t}$ and $Z_{t}$ and its Bayesian causal decomposition (see Section A.5 for derivation) as

(30)$$\begin{array}{cc} P_{S}\left[Y_{t},Z_{t}\right] & :=P_{S}\left[Z_{t}∥Y_{t}\right]Q\left[Y_{t}\right]=P_{S}^{B}\left[Y_{t}∥Z_{t}\right]P_{S}^{B}\left[Z_{t}∥Y_{t-1}\right]. \end{array}$$

In this decomposition,
(31)$$\begin{array}{cc} P_{S}^{B}\left[Y_{t}∥Z_{t}\right] & :=\prod_{\tau =0}^{t-1}T_{S}^{B}\left(y_{\tau +1}|z_{\tau +1},y_{\tau }\right)p\left(y_{0}|z_{0}\right), \end{array}$$
(32)$$\begin{array}{cc} P_{S}^{B}\left[Z_{t}∥Y_{t-1}\right] & :=\prod_{\tau =0}^{t-1}T_{S}^{B}\left(z_{\tau +1}|y_{\tau }\right)p\left(z_{0}\right) \end{array}$$
are path probabilities generated by two pairs of new transition probabilities that are obtained by Bayes’ theorem as
(33)$$\begin{array}{cc} T_{S}^{B}\left(z_{t+1}|Y_{t}\right) & :=\sum_{y_{t+1}}T_{S}\left(z_{t+1}|y_{t+1}\right)T_{E}^{F}\left(y_{t+1}|Y_{t}\right), \end{array}$$
(34)$$\begin{array}{cc} T_{S}^{B}\left(y_{t+1}|z_{t+1},Y_{t}\right) & :=\frac{T_{S}\left(z_{t+1}|y_{t+1}\right)T_{E}^{F}\left(y_{t+1}|Y_{t}\right)}{T_{S}^{B}\left(z_{t+1}|Y_{t}\right)}, \end{array}$$
where $T_{S}^{B}\left(y_{t+1}|z_{t+1},Y_{t}\right)$ is the Bayesian posterior of the environmental state, $y_{t+1}$, given the information of the sensed signal $z_{t+1}$ and the previous environmental state $y_{t}$. In this procedure, we switch the causal order of $y_{t}$ and $z_{t}$ by using Bayes’ theorem. Then, by using Equations (Equation 15) and (Equation 21) can be rearranged as
(35)$$\begin{array}{c} e^{-\left(\Psi^{i}\left[Y_{t}\right]-\left(\Psi_{0}\left[Y_{t}\right]+i\left[Z_{t}\to Y_{t}\right]+g\left[Y_{t},Z_{t}\right]\right)\right)}=\frac{P_{S}^{B}\left[Y_{t}∥Z_{t}\right]}{P_{K,F}\left[Y_{t}|Z_{t}\right]}, \end{array}$$
where $P_{K,F}\left[Y_{t}|Z_{t}\right]:=\sum_{X_{t}}P_{K}\left[Y_{t}∥X_{t}\right]P_{F}\left[X_{t}∥Z_{t}\right]$ and $i\left[Z_{t}\to Y_{t}\right]:=\ln P_{S}^{B}\left[Y_{t}∥Z_{t}\right]/Q\left[Y_{t}\right]$ is the pointwise directed information from $Z_{t}$ to $Y_{t}$ (see Section A.6 for derivation). This is another detailed FR with individual sensing, the average version of which can be obtained by taking the average with respect to $P_{S}\left[Y_{t},Z_{t}\right]$:(36)$$\begin{array}{c} {\langle \Psi^{i}\rangle }_{Q}={\langle \Psi_{0}\rangle }_{Q}+I^{Z_{t}\to Y_{t}}+G-D_{\mathrm{loss}}, \end{array}$$
where $D_{\mathrm{loss}}=D\left[P_{S}\left[Y_{t},Z_{t}\right]∥P_{K,F}\left[Y_{t}|Z_{t}\right]P_{S}^{B}\left[Z_{t}∥Y_{t-1}\right]\right]$ and $I^{Z_{t}\to Y_{t}}:={\langle i[Z_{t}\to Y_{t}]\rangle }_{P_{S}\left[Y_{t},Z_{t}\right]}$ is the directed information [31]. Directed information is an extension of mutual information of two trajectories by considering the causal relationship between the trajectories. Similarly to transfer entropy, the directed information quantifies the causal dependency between two trajectories. Transfer entropy is related to the upper bound of the rate of the directed information [35]. Their integral version is illustrated numerically in Figure 5e,f. Since $g\left[Y_{t},Z_{t}\right]=\Psi^{i}\left[Y_{t}\right]-\Psi^{c}\left[Y_{t},Z_{t}\right]$, we can immediately see that Equations (Equation 35) and (Equation 36) are equivalent to the detailed and average FRs, respectively, for the fitness with common sensing:(37)$$\begin{array}{c} e^{-\left(\Psi^{c}\left[Y_{t},Z_{t}\right]-\left(\Psi_{0}\left[Y_{t}\right]+i\left[Z_{t}\to Y_{t}\right]\right)\right)}=\frac{P_{S}^{B}\left[Y_{t}∥Z_{t}\right]}{P_{K,F}\left[Y_{t}|Z_{t}\right]}, \end{array}$$
and
(38)$$\begin{array}{c} {\langle \Psi^{c}\rangle }_{Q}={\langle \Psi_{0}\rangle }_{Q}+I^{Z_{t}\to Y_{t}}-D_{\mathrm{loss}}. \end{array}$$

These relations for the common sensing were originally derived in [27]. For a given and fixed sensing property, $T_{S}\left(z_{\tau }|y_{\tau }\right)$, the maximum gain of the average fitness by common sensing is shown to be bounded by $I^{Z_{t}\to Y_{t}}$ as
(39)$$\begin{array}{c} \max_{T_{F},T_{K}}{\langle \Psi^{c}\rangle }_{Q}-{\langle \Psi_{0}\rangle }_{Q}\leq I^{Z_{t}\to Y_{t}}, \end{array}$$
where the equality is attained when $D_{\mathrm{loss}}=0$. $D_{\mathrm{loss}}$ is the loss of fitness due to an imperfect implementation of a sequential Bayesian inference, and becomes 0 if and only if the phenotypic switching strategy, $P_{F}^{*}\left[X_{t}∥Z_{t}\right]$, and the metabolic allocation strategy, $P_{K}^{*}\left[Y_{t}∥X_{t}\right]$, are jointly optimized to implement the Bayesian sequential inference as $P_{K,F}^{*}\left[Y_{t}|Z_{t}\right]=P_{S}^{B}\left[Y_{t}∥Z_{t}\right]$, where
$$\begin{array}{cc} P_{K,F}^{*}\left[Y_{t}|Z_{t}\right] & :=\sum_{X_{t}}P_{K}^{*}\left[Y_{t}∥X_{t}\right]P_{F}^{*}\left[X_{t}∥Z_{t}\right]. \end{array}$$

An instance of the optimal metabolic allocation and phenotypic switching strategies is $T_{K}^{*}\left(y|x\right)=\delta_{x,y}$ and $T_{F}^{*}\left(x^{\prime }|x,z\right)={\left.T_{E}^{B}\left(y^{\prime }|z,y\right)\right|}_{y^{\prime }=x^{\prime },y=x}$, when $S^{x}=S^{y}$. These optimal strategies mean that, for each phenotypic state, all metabolic resources are allocated to one of the environmental states, and the phenotype switches with the probability that is exactly the same as the Bayesian posterior of the environment given the sensing signal. In other words, cells with phenotype *x* can survive and grow only when the environment is in the state to which all the metabolic resource is allocated under the phenotype *x*, and the cells change their phenotypic state by calculating the Bayesian posterior probability of the next environmental state given the common sensing signal.

In contrast, in the case of individual sensing, the Bayesian inference is no longer optimal, as G is dependent on the strategies of phenotypic switching and metabolic allocation, and $\left\{P_{F}^{*},P_{K}^{*}\right\}$ may not be the maximizer of G. This fact is more clearly shown as
(40)$$\begin{array}{c} \max_{T_{F},T_{K}}{\langle \Psi^{i}\rangle }_{Q}\geq {\langle \Psi^{i*}\rangle }_{Q}={\langle \Psi_{0}\rangle }_{Q}+I^{Z_{t}\to Y_{t}}+G^{*}, \end{array}$$
where $\Psi^{i*}$ and $G^{*}$ are obtained by inserting $P_{F}^{*}$ and $P_{K}^{*}$ that satisfy $D_{\mathrm{loss}}=0$. Equivalently, from $\sigma \left[Y_{t}\right]=\Psi^{i}\left[Y_{t}\right]-\Psi_{0}\left[Y_{t}\right]$, we have
(41)$$\begin{array}{c} {\langle \sigma [Y_{t}]\rangle }_{Q[Y_{t}]}=I^{Z_{t}\to Y_{t}}+G-D_{\mathrm{loss}}, \end{array}$$
and
(42)$$\begin{array}{c} \max_{T_{F},T_{K}}{\langle \sigma [Y_{t}]\rangle }_{Q[Y_{t}]}\geq I^{Z_{t}\to Y_{t}}+G^{*}. \end{array}$$

This inequality further indicates that the maximum average fitness gain from individual sensing for a fixed sensing strategy is greater than or equal to the directed information plus $G^{*}$, which means that the sequential Bayesian inference is no longer optimal. It is optimal in the case of the common sensing as the sensing signal is common and the subsequent phenotypic diversification by following the sequential Bayesian inference can hedge the risk of the error optimally. In the individual sensing, in contrast, stochastic individual sensing automatically induces a diversification in a population, which makes subsequent diversification by following Bayesian posterior suboptimal and redundant. Moreover, the information measure of the sensing, such as directed information, may not be an appropriate quantity to capture the efficiency of the overall decision-making process with individual sensing.

## 5. Discussion and Future Works

These results indicate that $\sigma [Y_{t}]$ and $\gamma_{t}$ are more relevant quantities for characterizing the fitness gain from the individual sensing. From the average FR of $\sigma [Y_{t}]$:$$\begin{array}{c} {\langle \sigma \rangle }_{Q}=\gamma_{t}-D\left[Q\left[Y_{t}\right]∥P_{\gamma }\left[Y_{t}\right]\right], \end{array}$$
the maximization of ${\langle \sigma \rangle }_{Q}$ is reduced to balancing the maximization of the total fidelity $\gamma_{t}$ and the minimization of $D[Q\left[Y_{t}\right]∥P_{\gamma }\left[Y_{t}\right]]$. As both $\gamma_{t}$ and $P_{\gamma }\left[Y_{t}\right]$ depend on the actual strategies of organisms, there exists a tradeoff between them, in general.

In the analogy of autoencoding and decoding, $\gamma_{t}$ becomes higher when each input $Y_{t}$ is decoded more correctly. In contrast, $D[Q\left[Y_{t}\right]∥P_{\gamma }\left[Y_{t}\right]]$ is minimized when the relative fidelity for $Y_{t}$ matches the probability, $Q[Y_{t}]$, that the environmental history $Y_{t}$ appears, since $P_{\gamma }\left[Y_{t}\right]$ measures the relative fidelity of decoding $Y_{t}$, given $Y_{t}$ as encoding information. From the definition of $P_{\gamma }\left[Y_{t}\right]$, Equation (Equation 24), $P_{\gamma }\left[Y_{t}\right]\leq e^{-\gamma_{t}}$ must hold for each $Y_{t}$. If the total fidelity $\gamma_{t}$ is fixed and small enough to satisfy $\max_{Y_{t}}Q\left[Y_{t}\right]\leq e^{-\gamma_{t}}$, balancing sensing, phenotypic switching, and metabolic allocation to satisfy $P_{\gamma }\left[Y_{t}\right]=Q\left[Y_{t}\right]$ becomes the optimal strategy to maximize $\langle \sigma \rangle$. This observation suggests that, under biologically realistic situations with moderate total fidelity, $P_{\gamma }\left[Y_{t}\right]=Q\left[Y_{t}\right]$ can be regarded as a proxy of the optimal strategy with individual sensing. If the total fidelity is too high to violate $\max_{Y_{t}}Q\left[Y_{t}\right]<e^{-\gamma_{t}}$, however, $D[Q\left[Y_{t}\right]∥P_{\gamma }\left[Y_{t}\right]]=0$ cannot be achieved, and more complicated optimization is required.

These investigations in conjunction with the analogy of the problem with autoencoding and decoding, show that in order to understand the decision-making of cells and organisms with individual sensing, we should consider a joint optimization of sensing, phenotypic switching, and metabolic allocation, rather than an optimization of a part of them with the other fixed and given [36]. In the evolution of cellular and organismal decision-making, these three factors are concurrently subject to natural selection, and we have to frame this problem appropriately. This challenge may lead to a deeper understanding of thermodynamics with feedback, because similar quantities to $\sigma [Y_{t}]$ and $\gamma_{t}$ have appeared already in the problem of feedback efficiency in information thermodynamics [33,34]. Moreover, the analogy of the problem with auto-encoding may pave the way to link the field of machine learning and deep learning with that of evolutionary biology and optimization.

Finally, we should note that all the information-theoretic relations derived in this work as well as others in previous works basically assume no cell interactions and no feedback from organisms to the environment. Even though extending the relations to relax such assumptions is a difficult problem, it can substantially expand the applicability of the information-theoretic approach to various biological problems.

## Figures and Tables

**Figure 1 entropy-21-01002-f001:**
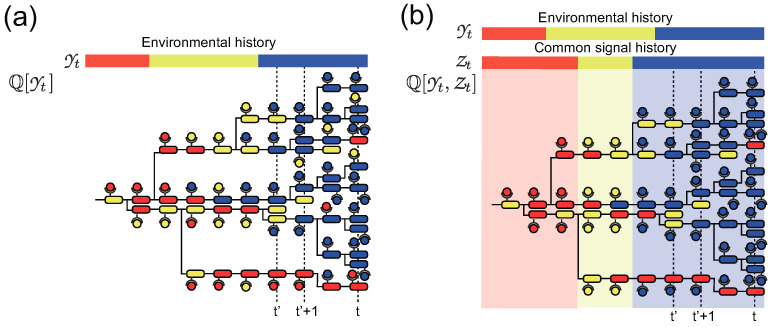
Schematic diagrams of population dynamics of cells with individual (**a**) and common (**b**) sensing. The colors of cells and molecules on the cells represent phenotypic states and sensing signal, respectively. Bars on the diagrams indicate the histories of environmental states and common sensing. In (**a**), the sensing singal of each cell is correlated with the environmental state but has an intercellular variation due to the stochasticity of individual sensing. In (**b**), on the other hand, all the cells at certain time points share the same sensing signal, which is shown by the background colors in the diagram.

**Figure 2 entropy-21-01002-f002:**
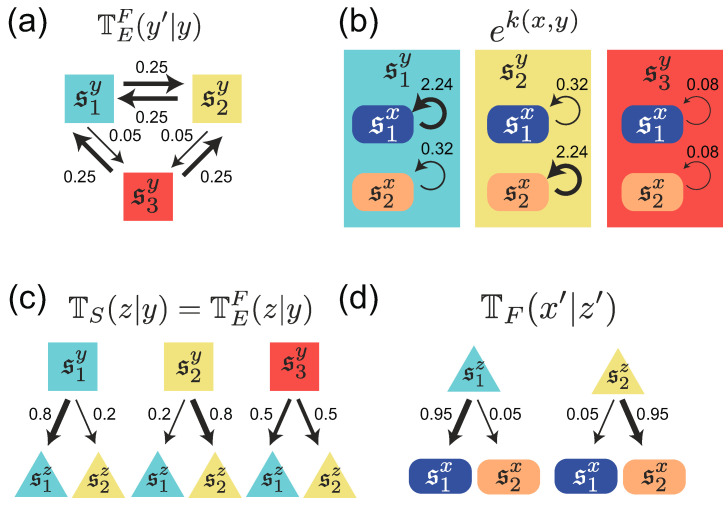
(**a**) A diagrammatic representation of state transitions of the environment used for simulation in Figure 3, Figure 4 and Figure 5. Three states are assumed for the environment; (**b**) Replication rates of cells with two different phenotypic states under different environmental states; (**c**) Environment-dependence of the sensing signal, and the probabilities to obtain a certain sensing signal under each environmental state; (**d**) Signal-dependent phenotype switching; The thickness of arrows represent relative probabilities and rates of replications. The values of the parameters used for the simulation are given by Equations (Equation 11)–(Equation 14).

**Figure 3 entropy-21-01002-f003:**
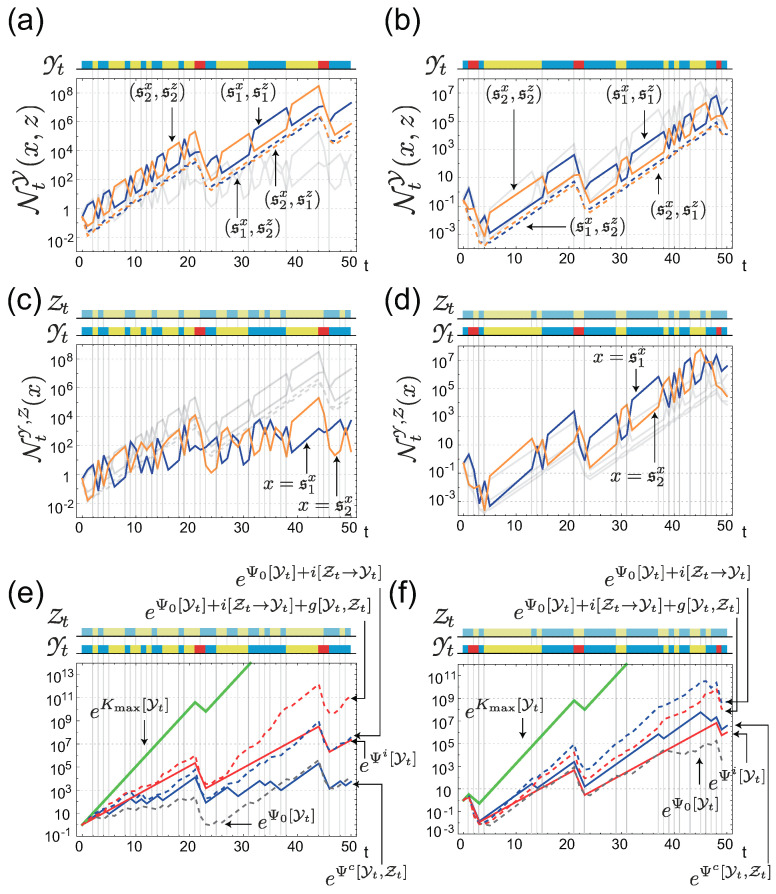
(**a**,**b**) Trajectories of population sizes with individual sensing under two different realizations of the environment. The history of the environment, $Y_{t}$, is shown by the colored bar on each panel. The color represents the state of the environment, which was defined in Figure 2a. Each colored line corresponds to the population size of the cells with phenotypic state *x* and sensing signal *z*; the actual value of (x,z) is designated in the panels. The gray lines in (**a**,**b**) are the replicate of the trajectories in (**c**,**d**), respectively, for comparison. (**c**,**d**) Trajectories of population sizes with common sensing under the same realizations of the environment as in (**a**) and (**b**), respectively. On each panel, the history of the common signal, $Z_{t}$, is additionally shown by the colored bar. Each colored line corresponds to the population size of the cells with phenotypic state *x*, with the actual value of *x* designated in the panels. (**e**,**f**) Fitnesses of the populations with the individual and the common sensing, $\Psi^{i}\left[Y_{t}\right]$ (red solid curve) and $\Psi^{c}\left[Y_{t},Z_{t}\right]$ (blue solid curve) under the same realizations of the environment and common signal as in (**a**,**c**) and (**b**,**d**). Related quantities are also shown for comparison.

**Figure 4 entropy-21-01002-f004:**
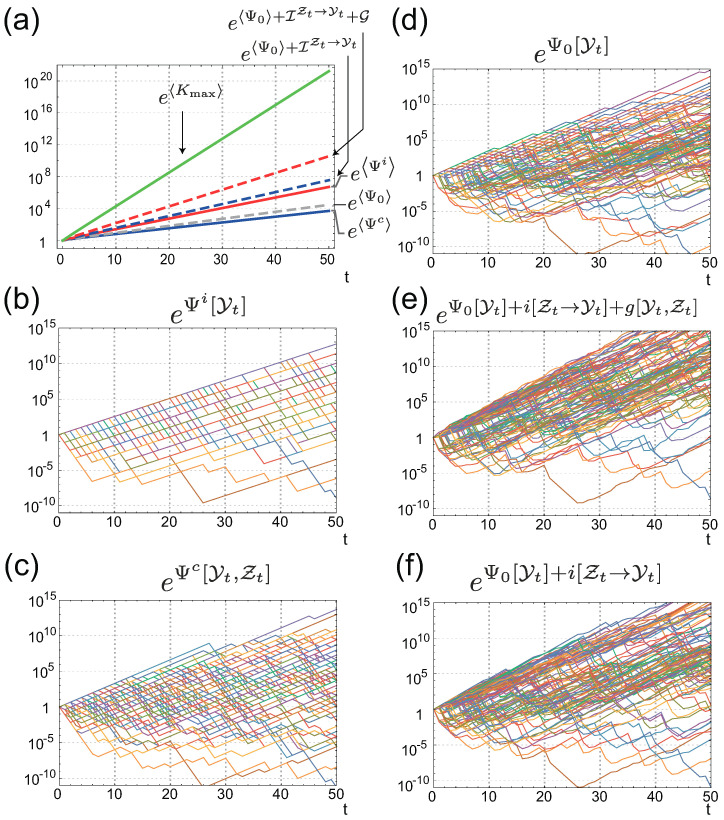
(**a**) Average values of fitnesses and related quantities; (**b**,**c**) Fluctuation of the fitness with individual sensing $\Psi^{i}\left[Y_{t}\right]$ (**b**); and that with common sensing $\Psi^{c}\left[Y_{t}\right]$ (**c**); (**d**–**f**) Fluctuation of $\Psi_{0}\left[Y_{t}\right]$ (**d**); $\Psi_{0}\left[Y_{t}\right]+i\left[Z_{t}\to Y_{t}\right]+g\left[Y_{t},Z_{t}\right]$ (**e**); and $\Psi_{0}\left[Y_{t}\right]+i\left[Z_{t}\to Y_{t}\right]$ (**f**).

**Figure 5 entropy-21-01002-f005:**
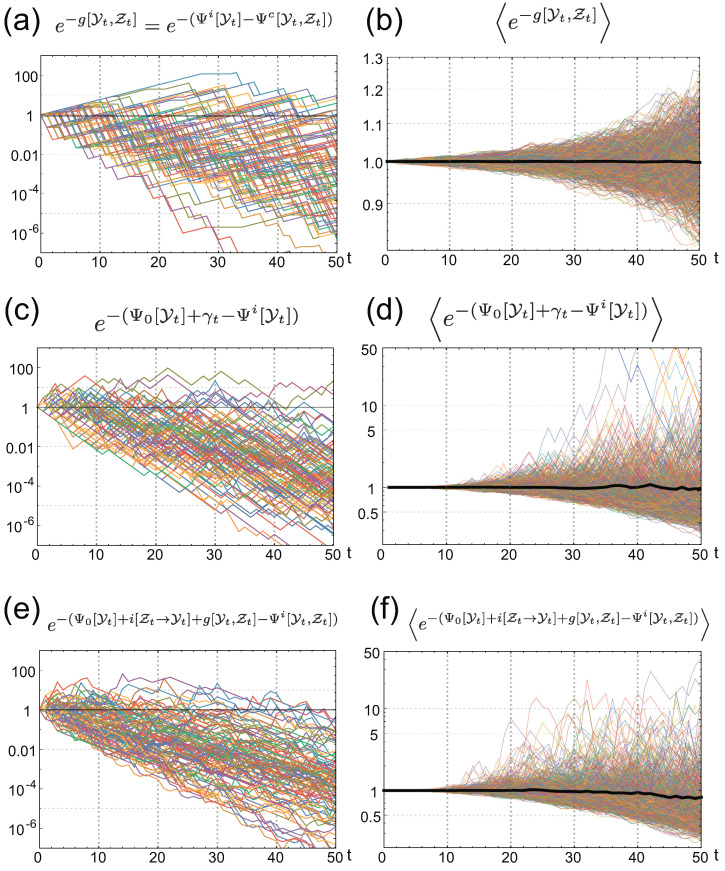
Numerical verification of IFRs for $g[Y_{t}]$ (**a**,**b**); $\gamma_{t}-\sigma \left[Y_{t}\right]$ (**c**,**d**); and $\Psi_{0}\left[Y_{t}\right]+i\left[Z_{t}\to Y_{t}\right]+g\left[Y_{t},Z_{t}\right]-\Psi^{i}\left[Y_{t}\right]$ (**e**,**f**). Left panels are behaviors of the integrands of the IFRs for 100 different realizations of the environmental and common signal histories. Right panels are the sample averages of the integrands of the IFRs. Thin colored curves are obtained by averaging $10^{5}$ different samples, and the thick black curves are obtained by the average of $1.2\times 10^{8}$ samples.

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
