# Peer review of "Fitness Gain of Individually Sensed Information by Cells"

_entropy, 2019, doi:10.3390/e21101002_

Round 1
Reviewer 1 Report
The paper proves fluctuation relations that provide detailed information on why the value of individual sensing is higher than that of collective sensing. The article seems to be original and scientifically sound. But I think that the readability of the paper may be improved further by modifying mathematical typos and supplementing derivations of some equations as follows:
1) In page 2, before equation (1) we may have T_S(z' | z, y) instead of T_S(z | y) for the sensing output.
2) In equation (5), I wonder whether we might have p_F(x_0 | z_0) instead of p_F(x_0).
3) In the expression for Psi^i[Y_t] between Eqs. (6) and (7), the average is taken concerning P_{F, S}[X_t|Y_t] which is defined by the summation over Z_t. But it seems that we should take the summation over all X_t and Z_t.
4) In Eqs. (27) and (28), we may have tau as the dummy index for the multiplication with y_tau and z_tau instead of t.
5) In pages 11-12, it may be not easy for the general readers of entropy to extract information from densely-written equations. Some additional references (e.g., for Bayesian causal decomposition) or helpful comments for the expressions may improve the readability of the paper.
Author Response
Thank you very much for your quick review and for providing valuable comments.
Here is the response to your comments.
In the revised manuscript, we designate our modification in red.
1) In page 2, before equation (1) we may have T_S(z' | z, y) instead of T_S(z | y) for the sensing output.
Thank you for pointing out. We corrected that.
2) In equation (5), I wonder whether we might have p_F(x_0 | z_0) instead of p_F(x_0).
Thank you again for pointing out. We corrected that.
3) In the expression for Psi^i[Y_t] between Eqs. (6) and (7), the average is taken concerning P_{F, S}[X_t|Y_t] which is defined by the summation over Z_t. But it seems that we should take the summation over all X_t and Z_t.
The summation over X is represented by the average <>. To clarify the more detail, we added Appendix A.1 for derivation.
4) In Eqs. (27) and (28), we may have tau as the dummy index for the multiplication with y_tau and z_tau instead of t.
We overlooked this mistake. We appreciate your suggestion.
5) In pages 11-12, it may be not easy for the general readers of entropy to extract information from densely-written equations. Some additional references (e.g., for Bayesian causal decomposition) or helpful comments for the expressions may improve the readability of the paper.
We are sorry for our insufficient explanation about this new term. We added more explanation and how we can derive the Bayesian causal decomposition in Appendix A.5.
Reviewer 2 Report
In “Fitness Gain by Individually Senses Information by Cells”, the authors construct a model to describe a model of sensing and adaptation and then investigate two sensing strategies, individual and common, by their relative fitness gains. While the paper derives some very interesting relationships, I have many concerns.
First, the authors utilize an exponential growth model, whereas it is most common to use a logistic growth model in which the population has a carrying capacity. It would be expected that the carrying capacity would be both dependent on the environment (based on the total amount of available resources) and the relative fraction of cells of each phenotype (based on the resource consumption of each phenotypic state). The existence of a carrying capacity would couple individual cells to the greater population and likely have an effect on the difference between the two sensing models.
Second, the authors use the transition probabilities T(x_t+1 | x_t, z_t+1) and T(z_t+1 | z_t, y_t+1). I believe these transition probabilities should be T(x_t+1 | x_t, z_t) and T(z_t+1 | z_t, y_t). Consider an actual sensing/adapting process, such as how the Trp repressor senses the concentration of available L-Trp and then represses the expression of Trp biosynthesis genes if the concentration is high. Y is the L-Trp concentration, Z is the state of the Trp repressor (active or inactive) and X in whether or not a gene under the Trp operon is repressed. In any sensible kinetic model of the system, the probability of a Trp repressor being in the active state at time t + 1 is related to its state at time t and the concentration of L-Trp at time t, not time t + 1. Similarly, the probability of the gene being repressed at time t + 1 (which occurs due to the activated Trp repressor binding the Trp operon) is related to whether the gene is already repressed at time t and the whether the Trp repressor is activated at time t. It is unclear how the instantaneous model affects the outcome, but it likely not representative of a true biochemical sensing/adapting process.
Also one expects additional feedback in the X - > Y direction. The authors use the example of an organism that changes its state based on the available nutrients, In this case the organism consumes the environment, and thus one would imagine that a T(y_t | y, z) term is necessary.
Finally, the authors investigate the directed information, but not the transfer entropy, which has been suggested to be its upper bound. A comment on transfer entropy should be made.
Author Response
Thank you very much for your quick review and for providing valuable comments.
Here is the response to your comments.
In the revised manuscript, we designate our modification in red.
First, the authors utilize an exponential growth model, whereas it is most common to use a logistic growth model in which the population has a carrying capacity. It would be expected that the carrying capacity would be both dependent on the environment (based on the total amount of available resources) and the relative fraction of cells of each phenotype (based on the resource consumption of each phenotypic state). The existence of a carrying capacity would couple individual cells to the greater population and likely have an effect on the difference between the two sensing models.
We agree that the carrying capacity is an important factor when we consider biologically general situations.
Even with this importance, at the same time, the carrying capacity makes the model of population dynamics more complicated and less tractable analytically.
This prevents us from directly relating the population dynamics and selection with information-theoretic quantities analytically.
At this moment, as long as we know, no work succeed in such an extension.
(We should note that numerical calculations of information measures by simulations is still possible, but this does not provide any information about possible bound and constraint of fitness in terms of information measures.)
In the revised manuscript, we explicitly mentioned this limitation in the introduction and added this topic as a future challenge in Discussion.
Second, the authors use the transition probabilities T(x_t+1 | x_t, z_t+1) and T(z_t+1 | z_t, y_t+1). I believe these transition probabilities should be T(x_t+1 | x_t, z_t) and T(z_t+1 | z_t, y_t).
Consider an actual sensing/adapting process, such as how the Trp repressor senses the concentration of available L-Trp and then represses the expression of Trp biosynthesis genes if the concentration is high. Y is the L-Trp concentration, Z is the state of the Trp repressor (active or inactive) and X in whether or not a gene under the Trp operon is repressed. In any sensible kinetic model of the system, the probability of a Trp repressor being in the active state at time t + 1 is related to its state at time t and the concentration of L-Trp at time t, not time t + 1.
Similarly, the probability of the gene being repressed at time t + 1 (which occurs due to the activated Trp repressor binding the Trp operon) is related to whether the gene is already repressed at time t and the whether the Trp repressor is activated at time t. It is unclear how the instantaneous model affects the outcome, but it likely not representative of a true biochemical sensing/adapting process.
Thank you for a comment on the time labelling in our model.
Your concern is related to the problem of how to order the causal relationship of multiple variables in the time discrete model.
In our model, we implicitly assume the following causal cascade (time-order):
y_t -> z_t -> x_t -> y_t+1 -> z_t+1 -> x_t+1
If we use a more accurate but more messy notation, they should be described as
y_t -> z_{t + 1/3} -> x_{t+2/3} -> y_{t+1} -> z_{t+1+1/3} -> x_{t+1+2/3}
Thereby, our T(x_t+1 | x_t, z_{t+1}) means in this notation that T(x_{t+1+2/3} | x_{t+2/3}, z_{t+1/3})
This actually is consistent with what you exemplified your concern with Top system.
The same true for T(z_t+1 | z_t, y_t).
In the revised manuscript, we explicitly mentioned this convention without confusing the readers.
Also one expects additional feedback in the X - > Y direction. The authors use the example of an organism that changes its state based on the available nutrients, In this case the organism consumes the environment, and thus one would imagine that a T(y_t | y, z) term is necessary.
The feedback from cells or organisms to the environment is actually a very important and quite interesting factor.
But similarly to the case of the carrying capacity, it is still very difficult and totally unsolved to involve this effect
While we are capturing the relationship between population dynamics and information measures.
No work succeed in such extension as well.
Therefore, in the revised manuscript, we explicitly mentioned this limitation in the introduction and added this topic as a future challenge in Discussion.
Finally, the authors investigate the directed information, but not the transfer entropy, which has been suggested to be its upper bound. A comment on transfer entropy should be made.
Thank you for the suggestion.
In the revised manuscript, we included an explanation about the connection of directed information to transfer entropy.
Reviewer 3 Report
In their paper, the authors study the gain in fitness that populations have when a stochastically fluctuating environment is sensed individually rather than by the population as a whole. More precisely, they compare expected population sizes and find that individual sensing leads to larger sizes because cells that realize a good sensing of the environment by chance get amplified in the population due to the larger growth rates that are realized by those cells. It is shown that the difference of expected population sizes is determined by the difference between the "forward probability distribution" of sensing and the backward probability that the sensing is realized when the difference in growth rates that it implies in individual sensing is taken into account. I am not too familiar with previous work on that topic or with studies of fitness/evolution in general and cannot judge too well how novel and important the results are for the authors' field. I can say, however, that for me personally the paper was quite educational. Furthermore, for the biological problems that I know, investigating the selection consequences of a stochastic process inside individual cells that non-genetically affects cellular growth rates is an important but mathematically underexplored question, to the point that most people probably do not even realize that this is a question that exists. Hence, I think that the author's paper is very timely and possibly also relevant beyond what is claimed as applications in the paper. I recommend publication and have the following suggestions for how the paper could still be improved.
(i) It is unnecessarily difficult to follow all the equations in the paper. For instance, while Eq.13 is rather trivial, realizing that is trivial and why is not so trivial in the way in which it is presented in the paper. For most parts of the paper, I would have wished that I could just click on the equality signs to expand the equation with the hidden steps instead of having to redo the steps myself to see why the equation is true. I realize that the authors do not want to clutter their paper with derivations containing lots of trivial steps but maybe an appendix or a better text-based explanation would make it easier for the reader to follow all the equality signs.
(ii) The figure captions are rather short and do not really explain all the content that is shown in the figures. Furthermore, I would suggest to change the thin lines in some of the figures to something that is a bit better visible.
(iii) In general, the considered biological context in the paper makes a lot of sense to me. But then, in Section 4.1, I lost the biological understanding a bit. Why is the phenotype x not already incorporating how metabolic resources are allocated? Or should I rather interpret the allocation as an explanation of the phenotypes rather than as something that cells do in addition to choosing a phenotype? What is the biological motivation for assuming that resources that are allocated to one type of environment are not contributing anything to growth in a different environment? I think in addition to the mathematical assumptions, being clearer on the biological motivation would be useful for this section.
And finally, I have some what if questions to the authors but I do not necessarily expect an answer.
What if time would be taken as continuous instead of discrete?
What if growth was not exponential?
What if the state/quality of the environment would depend on what the cells are doing?
What could be done to validate the theory experimentally?
Author Response
Thank you very much for your quick review and for providing valuable comments.
Here is the response to your comments.
In the revised manuscript, we designate our modification in red.
(i) It is unnecessarily difficult to follow all the equations in the paper.
For instance, while Eq.13 is rather trivial, realizing that is trivial and why is not so trivial in the way in which it is presented in the paper.
For most parts of the paper, I would have wished that I could just click on the equality signs to expand the equation with the hidden steps instead of having to redo the steps myself to see why the equation is true. I realize that the authors do not want to clutter their paper with derivations containing lots of trivial steps but maybe an appendix or a better text-based explanation would make it easier for the reader to follow all the equality signs.
Thank you so much for a very constructing comment. We agree that we definitely should include more details of how we derived some equations
to facilitate the readers to grasp more easily the concepts and results of our paper.
In the revised manuscript, by following your suggestion, we included Appendix in which we described how we derived important equations in the main text.
We also added more explanations to several equations and equalities.
(ii) The figure captions are rather short and do not really explain all the content that is shown in the figures. Furthermore, I would suggest to change the thin lines in some of the figures to something that is a bit better visible.
Thank you for your comment. We modified our manuscript, accordingly.
(iii) In general, the considered biological context in the paper makes a lot of sense to me. But then, in Section 4.1, I lost the biological understanding a bit. Why is the phenotype x not already incorporating how metabolic resources are allocated? Or should I rather interpret the allocation as an explanation of the phenotypes rather than as something that cells do in addition to choosing a phenotype? What is the biological motivation for assuming that resources that are allocated to one type of environment are not contributing anything to growth in a different environment? I think in addition to the mathematical assumptions, being clearer on the biological motivation would be useful for this section.
Thank you for your comment. The metabolic allocation is a way to characterize phenotypic states.
The allocation of each phenotypic state is fixed and therefore cells are just choosing phenotypic state.
We added more explanation about metabolic allocation and its biological meaning in the revised manuscript.
And finally, I have some what if questions to the authors but I do not
necessarily expect an answer.
What if time would be taken as continuous instead of discrete?
All the results can hold even for the continuous model.
Simple continuous limit of this discrete model leads to the situation that the interval of cell division is exponentially distributed, which is not biologically realistic.
Even if we consider other inter-division intervals such as gamma distribution, our results hold because all the details in the model are absorbed by the path-wise description.
We already employed such property to derive a response relation of fitness for a time-continuous branching population model without environmental fluctuation and sensing.
(https://journals.aps.org/pre/abstract/10.1103/PhysRevE.99.012413)
We did not touch on that because the detailed continuous model may prevent readers from understanding the essence of our results about information-theoretic aspects of population dynamics.
What if growth was not exponential?
We thank that we may be able to extend the results by introducing other information measures based not on KL divergence but on Renyi or alpha divergence,
Because the exponential growth is tightly linked to the KL divergence (an average of log ratio) in our results.
But we have not worked on that yet.
What if the state/quality of the environment would depend on what the
cells are doing?
That is a quite interesting extension. In such a case, we should consider a feedback relation between environment and cells.
Because reviewer 2 asked basically the same question, we added some note on such extension in Discussion.
We want to work on that topic in the near future.
What could be done to validate the theory experimentally?
To validate the theory, we should measure observables along with history of the cellular phenotypic state, because our results are heavily dependent on the pathwise formulation.
To this end, we are already working with an experimentalist who devised a new measurement system with which we can observe cells over 100 generations in a well-controlled condition. By using our theory on the continuous-time branching population, we also proposed a method with which we can infer hidden phenotypic states of cells based on cellular lineage trees obtained the experimental device under constant environmental condition (https://www.biorxiv.org/content/10.1101/488981v1).
We are now planning to extend this system to investigate the behaviour of cells in response to the changes in the environment.
At the same time, a theoretical task is to derive quantities that are linked to our information measures (e.g., directed information) and are observed more easily with a much smaller number of samples, because the size of experimental data is limited.
We are thinking about addressing this problem by employing the contraction technique in large deviation theory.
Round 2
Reviewer 1 Report
The authors have improved the readability of the manuscript greatly by providing additional comments and derivations of proofs. The paper is worth publishing in Entropy.
Reviewer 3 Report
I have no more comments.